# Evaluation of Brace Treatment Using the Soft Brace Spinaposture: A Four-Years Follow-Up

**DOI:** 10.3390/jcm11010264

**Published:** 2022-01-05

**Authors:** Christian Wong, Thomas B. Andersen

**Affiliations:** 1Department of Orthopedics, Hvidovre Hospital, Kettegaards Alle 30, 2650 Hvidovre, Denmark; 2Department of Orthopedics, National University Hospital, Blegdamsvej 9, 2100 Copenhagen, Denmark; thomas.borbjerg.andersen@regionh.dk

**Keywords:** spinal deformities, idiopathic scoliosis, bracing, pathobiomechanics, follow-up study

## Abstract

The braces of today are constructed to correct the frontal plane deformity of idiopathic adolescent scoliosis (AIS). The Spinaposture brace© (Spinaposture Aps, Copenhagen, Denmark) is a soft-fabric brace for AIS and is designed to enhance rotational axial stability by inducing a sagittal plane kyphotic correction. This prospective observational study evaluated the brace in fifteen patients with AIS. The initial average CA was 16.8° (SD: 2.8). They were followed prospectively every 3 to 6 months during their brace usage until skeletal maturity of 25 months and at long-term follow-up of 44 months. In- and out-of-brace radiographs were performed in six subjects at inclusion. This resulted in an immediate in-brace correction of 25.3 percent in CA (14.3°→10.8°) and induced a kyphotic effect of 14.9 percent (40.8°→47.9°). The average in-brace improvement at first follow-up was 4.5° in CA, and the CA at skeletal maturity was 11° (SD: 7.4°) and long-term 12.0° (SD: 6.8°). In conclusion, the Spinaposture brace© had an immediate in-brace deformity correction and a thoracic kyphotic effect. At skeletal maturity, the deformities improved more than expected when compared to that of the natural history/observation and similar to that of other soft braces. No long-term deformity progression was seen. To substantiate these findings, stronger designed studies with additional subjects are needed.

## 1. Introduction

Idiopathic adolescent scoliosis (AIS) is a structural spinal disorder that sometimes requires the intervention of either bracing or surgery due to the magnitude of or progression in the spinal deformity [1,2]. The bracing of today has a governing principle of three points pressure-mediated correction in the frontal plane supplemented by derotation to prevent curve progression using a hard brace of polyethene [1,2,3,4,5,6]. Such hard braces can have a negative psychological impact, which subsequently leads to decreased compliance and thereby diminished correctional effect [3,7,8]. The efficacy of the current treatment of bracing has been debated [9,10,11]. It has been claimed that omitting bracing is inconsequential [11]. However, the efficacy of bracing was tested in a rigorous scientifically design, and it minimizes the need for corrective surgery [3]. Doubts still loom concerning the efficacy of bracing despite the meticulous scientific effort [9]. Earlier studies have introduced soft braces, which consist of, for example, correctional elastic ribbons [12]. This seems to have a correctional effect, but it is not as effective as hard bracing (especially not in the period of pubertal growth) [12,13]. An insufficient correction in the sagittal profile in the first soft braces has been considered as a reason for the lack of correction [14]. Bracing concepts have now evolved to include not only restoring the frontal plane deformity but also focusing on correction in the sagittal alignment with seemingly promising results [12,15,16]. The Spinaposture© brace is a soft brace focused on sagittal alignment which consists of two parts. The first part is a soft ‘suit’ that is custom made to fit the individual anthropometric measures using the measures from a whole trunk optical 3D scan of the individual patient. The brace is sewn using Sensotex, a special elastic fabric of Neopren and Sensotex Lycra, and has a ‘memory effect’ (with the ability to return to its original shape). The brace is designed to induce a kyphotic effect. This is induced both by the tactile response to the elastic component of the fabric and incorporated into the design of the ‘suit’. The brace has a second part, namely a hard shield with ‘fingers’ to be placed in a ‘pocket’ of the soft fabric placed posterior at the thoracic level. This induces a further kyphotic effect. The hard shield is introduced secondarily if the frontal plane deformity is progressing (increased CA) or if initial correction is insufficient. Figure 1 displays the ‘suit’ and the ‘hard shield’ of the Spinaposture brace©.

The brace is intended to augment axial stability by inducing a sagittal plane thoracic kyphotic effect and, if needed, a derotation in the spine depending on the individual curve pattern. The brace is intentionally built to induce minor to no direct frontal plane correction, and to realign or restore the thoracic kyphosis by inducing an inherently proprioceptive ‘hyper-kyphotic strive’. The strive is induced utilizing the principle of autocorrection derived from physiotherapeutic practice [17]. Specifically, the aim is to realign the temporary, naturally occurring thoracic hyperkyphosis occurring during the normal growth in adolescence [18]. This is considered to have a role in the initial development of thoracic AIS [19]. AIS is typically associated with the adolescent female. AIS has been perceived as developing as a consequence of the spine growing into a mechanically unstable column and might progress in a self-sustaining ‘vicious cycle’ due to the effect of gravity and asymmetric loads in a growth-modulating buckling-like manner. The spinal growth spurt for both sexes coincides with adolescence, but adolescent females distinguish themselves from males by a significantly increased and earlier thoracic growth. These factors coincide with a temporal sagittal flattening of thoracic vertebrae for adolescent females. This contributes to rotational instability of the spine. The brace is aimed to improve or reestablish axial stability by the sagittal plane correction/realignment. The intended purpose is to prevent curve progression without being as strenuous as the conventional hard braces. The brace is designed as an advanced t-shirt/body stocking, and the discomfort of the brace is closer to regular clothing than a hard brace, which we perceive as beneficial for promoting compliance and subsequent treatment efficacy [8]. Being less strenuous whilst probably having a less frontal plane correctional effect, justified bracing smaller curves, and early intervention also has a better correctional effect [20]. Thus, we braced AIS with Cobb’s angle (CA) between 15° and 25° [21]. This study evaluated the correctional effect of the Spinaposture brace© with in- and out-of-brace radiographs for patients with AIS, and in a prospective follow-up study, we evaluated the initial correction when the brace was discontinued at skeletal maturity and long-term at approximately four years.

## 2. Materials and Methods

### 2.1. Inclusion, Bracing and Strengthening Exercises 

The subjects were patients referred with AIS from our hospital service area and included as a convenience sample. At the first visit, the including doctor diagnosed them with AIS, and if having a thoracic or thoracolumbar AIS with a CA between 15° and 25°, they were included after informed consent. The subjects and caregivers were specifically told that longitudinal radiological follow-up without bracing was the ‘gold standard’ treatment. Exclusion criteria were subjects with primary lumbar scoliosis, previous brace treatment or inability to cope with bracing in general, or if MRI-verified intraspinal pathology was present. If the subjects had a correction of approximately one-fourth in the frontal plane deformity after three months, when using the brace, this was continued. If the correction was less than one-fourth then the ‘hard shield’ was introduced. If the subjects had a correction of approximately one-fourth in the frontal plane deformity after another three months, when using the brace, then this was continued. Otherwise, the subjects were excluded. The ‘body stocking’ part of the brace was worn in the daytime (<16 h and defined as part-time) to counteract the effect of gravity on the spine and was not worn at night [13,22]. We allowed them not to wear the brace when performing sports and when the subjects were too hot in the brace. We considered the brace as a passive correctional measure [19], thus we advised them to perform strengthening exercises either through a home exercise program, formal physiotherapy and/or participate in sports such as crawl swimming [23]. We allowed all forms of physical exercises at the patient’s discretion throughout the study.

### 2.2. Initial in-Brace Radiographs 

At study initiation, the subjects were asked to volunteer to have performed initial in-brace radiographs (both anteroposterior and lateral projections). 

### 2.3. Outpatient Follow-Up

The subjects were followed prospectively with anteroposterior out-of-brace radiographs every six months in general. However, the initial radiographs were taken every three months as a precaution since we had no previous experience using the brace [12]. The subjects were instructed not to wear the brace one day before the follow-up. We used a local developed low-dose anteroposterior radiological examination [24]. Figure 2 shows a typical radiographic follow-up using the low dose radiographic technique. 

The brace was worn until skeletal maturity. Skeletal maturity was determined by having at least Risser 4, duration of menstruation of 2 years for girls, and a skeletal hand age of 14 years and 16 years for girls and boys (Sanders stage 8), respectively. The follow-up was continued for an additional three to six months with a final radiographical and clinical evaluation. The subjects were recalled for one long-term radiographic follow-up. 

### 2.4. Study Assessments and Statistical Analyses

The primary outcome was Cobb’s angle for the primary curves at skeletal maturity. A change of five degrees or more was considered either a progression or regression [13,25]. If the CA remained within five degrees of the initial measurement before the brace treatment was initiated, we considered them as ‘stable’. We compared the course of the spinal deformity when using the brace with comparison to the expected outcomes in relation to the Risser classification at brace initiation [13]. At skeletal maturity, 42 percent of AIS is expected to improve or remain stable for Risser 0–1 and 71 percent of AIS is expected to improve or remain stable for Risser 0–2 [13]. In this study, we considered an improvement or remaining stable as a good outcome. For observation only, the expected outcome is 50% (CI: 44–56%) [13].

Secondary parameters were radiographic and evaluated as categorical variables of changed or unchanged. The parameters were the descriptive categorization of scoliosis according to the Moe-Kettleson classification [26], changes in the level of the apex vertebrae (if more than two levels difference), Nash and Moe’s classification (if rotation changed more than one segment) at the apex vertebrae, change in rib vertebrae angle differences/Metha angle (if more than 20° difference), and change in CA of the secondary curves if any. 

The evaluators were two doctors of pediatric orthopaedic and spinal surgery with 26 and 20 years of experience, respectively. All measurements were performed separately and blinded using the Synedra View Personal 19 (Ver. 19.0.0.2, Innsbruck, Austria) and the PACS system (Impax 6.4.0, Agfa^®^ HealthCare, Mortsel, Belgium) on a three-megapixel viewing station. For statistical analyses, we performed an inter-class correlation for inter-observer variability between the evaluators. The chi-square cross-tabulation test and the Spearman correlation test were performed for differences between subjects and drop-outs. All tests were performed using IBM SPSS Statistics, Version 25 (IBM, Richmond, VA, USA). A post hoc power analysis was performed using GPower (Ver 3.0.10, Aichach, Germany).

The study was conducted according to the guidelines of the Declaration of Helsinki II. The local ethical committee evaluated this study as a clinical observational follow-up series (reference number H-17014162). Informed consent was obtained from all involved subjects. This research received no external funding.

## 3. Results

### 3.1. Population

Twenty-two patients were screened. Seven patients were excluded from the study at first evaluation. Two subjects had CA larger than 25° and previous brace treatment, three subjects were not included due to syrinx or anisomelia, and two subjects received the brace but never used it. Fifteen subjects were included with no prior treatment, except for observation. Five subjects dropped out. Two subjects, after joining a formal Schroth therapy, acquired the Chêneau Gensigen brace by their own accord. This was encouraged by their private physiotherapists. One subject had back pain without improvement by bracing, one changed to Rolfing structural integration therapy, and one did not attend the follow-ups. All three subjects discontinued bracing. Two subjects did not attend the long-term follow-up. One subject was too busy with school work and we were unable to recall one. We considered them as drop-outs. Otherwise, there were no subjects lost in follow-up or withdrawals. Figure 3 shows a flow chart of the history of the subject’s participation, exclusion, and drop-out.

For the ten subjects at skeletal maturity in the follow-up study, the average age at brace initiation was 12.26 (SD: 2.76) years. The clinical characteristics of the subjects and relevant medical history are summarized in Table 1. 

### 3.2. In-Brace Correction

Six subjects were examined for the in-brace correction with radiographs in the anteroposterior and lateral projections. The frontal plane correction and kyphotic effect were measured by comparing the CA measured at the same levels on the in- and out-of-bracing radiographs. The in-brace correction of the primary curve in the frontal plane was 25.3 percent (14.3° to 10.8°). The in-brace thoracic kyphosis in the sagittal plane increased 14.9 percent (40.8° to 47.9°). Figure 4 shows an example of the radiographic in-brace correction.

### 3.3. Follow-Up Study

In the prospective follow-up, the average CA when starting bracing was 16.8° (SD: 2.8°). The average correction in CA at the first follow-up was 4.5° (SD: 3.2°). Initially, four subjects (4/15) improved in CA with more than 5 dg. and eleven (11/15) remained stable. The average bracing period was 21 months (range: 12.4–37.63 months; SD: 9.5 months). The average CA at the end of bracing was 11.0° (SD: 7.2°). At end of bracing, six subjects (6/10) had straight spines with a CA less than ten degrees and four (4/10) remained stable. In accordance with Costa et al. (2021), the expected outcome for bracing at skeletal maturity was 6/11 and 9/15 for the Risser stage 0–1 and 0–2 at brace initiation, respectively [13]. The achieved outcome was 11/11 and 15/15 for the Risser stage 0–1 and 0–2, respectively, when using the brace. The expected outcome for observation is 50% (CI 44–56%) [13]. Thus, approximately 8/15 was expected to improve or be stable by observation/natural history. In the long term, none of the subjects deteriorated. When extrapolating the expected outcome from Costa et al. (2021), the expected outcome for bracing long-term was 4/9 and 7/10 for the Risser stage 0–1 and 0–2 at brace initiation, respectively [13]. The achieved outcome was 9/9 and 10/150 for the Risser stage 0–1 and 0–2, respectively, when using the brace. The last follow-up was at 24 months (SD: 7.5 months). The average CA at the last follow-up was 10.0° (SD: 6.8). The average long-term follow-up was at 44 months (SD: 9.3 months). The average CA at the long-term follow-up was 12.0° (SD: 6.8°). Three subjects (3/8) had straight spines and five (5/8) remained stable. The individual curve developments are shown in the Appendix A. Figure 5 shows the average changes over time (left-A) and Table 2 displays the achieved and expected outcomes (right-B). 

Secondary radiographic parameters in the follow-up study were evaluated according to if changed and are illustrated in Table 3.

### 3.4. Brace Compliance and Events

The compliance of brace usage and events were evaluated by open questioning at every follow-up where the subjects and caregivers were encouraged to disclose non-compliance and discomfort when using the brace in an open-minded dialogue. Non-compliance and brace events were seen in periods of warm weather and the subjects often had subsequently deformity progression at the next follow-up. This motivated the subjects to a more rigorous brace usage motivated by the deterioration (see Appendix A). Two subjects had heat sensors embedded in their braces. This is considered an objective way of measuring brace wear compliance by body heat [27]. The heat sensor would register an increase in temperature when the brace was worn. The compliance for those two subjects seemed to be fulfilled. Figure 6 (Upper-A) shows an example of the monitoring with spikes of increased temperature when the brace was worn, and Table 4 (Lower-B) illustrates the events and bracing history.

When performing analysis for interobserver variation, the two doctors had an excellent inter-class correlation of 0.85 [25].

When performing a posthoc power analysis, the power (1-β, α = 0.05) for the first follow-up with 15 subjects was 0.98, at skeletal maturity 0.88 for ten subjects and long-term 0.79 for eight subjects. 

When performing the Chi-square cross-tabulation, none of the parameters of gender, age, Risser stage, and initial CA between subjects and dropouts was statistically significant (z < 1.96). The Spearman correlation for dropout found a positive and negative association between initial CA (*p* < 0.001) and the duration of bracing (*p* < 0.01), respectively.

## 4. Discussion

In this study, we examined a new brace for adolescent idiopathic scoliosis. When testing for immediate correction by in- and out-of-brace radiographs, the proprioceptive-mediated ‘strive’ of the brace induced a thoracic sagittal kyphotic effect. This led to an immediate thoracic frontal plane correction of the AIS deformity. This suggests that there can be an interrelationship between the frontal and sagittal plane spinal curves of the spine which is described as ‘coupled motions’ by Panjabi and White [28]. To our knowledge, this has not been demonstrated in vivo in humans. In the follow-up study, there was a deformity correction of one-fourth at the first follow-up visit. At skeletal maturity, all subjects either improved or remained stable in their frontal plane deformity. When compared to the expected outcome for observation only (50% success rate) or other soft braces (62.5% success rate) [13], more subjects were improving or remaining stable when using the soft brace when stratified to the Risser stage 0–1 and 0–2 at brace initiation. In the long term, none of the subjects deteriorated. Based on these findings, the correctional effect or brace efficacy for this soft brace was better than those of previous studies with soft bracing and observation only. One can argue that more subjects are needed to detect an effect (especially long-term) but as a precautious measure, only a small cohort of subjects was included. In conclusion, our results for the Spinaposture brace are comparable to those of previous studies with soft braces when used part-time and for small curve AIS.

When interpreting our results, we had the following deliberations. In the initial evaluations of the subjects, we performed in- and out-of-brace radiographs and closer follow-ups with three months intervals in the initial period of brace wearing as well. The in and out-of-brace radiographs were voluntary for the subjects since this induced additional radiation [29]. We refrained from in and out-of-brace radiographs throughout the study for this reason. The initial closer follow-up was carried out as a precautious measure. We justified the more frequent follow-up and excess radiation [30] by using a local developed low dose radiographic technique [24]. This exposed the subjects to eight-fold lower radiation doses than the standard posteroanterior radiographs. In this study, only a small cohort of subjects was included as a precautious measure when testing a new brace, and since this study was a clinical follow-up study, we performed a posthoc power analysis that determined our findings as adequately powered at skeletal maturity. However, we acknowledge the impediments of posthoc power analysis [31]. Our subjects were chosen as a convenience sample, and thus our study could be influenced by selection bias. However, there were no differences in demography of gender and age distribution when compared to previous studies [3,32], thus indicating that our subjects were comparable. Our study could also be influenced by selection bias due to our exclusions and drop-outs. Five of the seven initial exclusions were due to the encouragement of private therapists using different braces and exercise therapies despite neither of the subjects progressing in deformity before converting to other treatments motivating this change. Non-compliance to any spine-related intervention was another cause of exclusion and the subsequent drop-outs during the study were either due to being unable to contact them or being too busy to attend the follow-up. We were unable to identify specific parameters that caused the drop-outs since there were no significant differences between subjects and dropouts when comparing the parameters. However, we found an association between the initial corrective effect of the brace and the duration of brace treatment and dropout. 

The purpose of examining the brace was two-fold, namely to discover if the hypnotized frontal plane deformity would be correctable using a sagittal plane strive and to determine if the brace was effective at skeletal maturity and long-term. The hard braces used today are strenuous, causing pressure marks, pain/soreness, and general discomfort with low compliance to follow [2,7,8]. The Spinaposture brace© was designed to be less strenuous and to be a supplement to our current non-operative treatment of watchful follow-up and physiotherapy. We acknowledge that using temperature monitoring for all subjects are needed for adequate and quantified brace compliance evaluation [3]. However, this first study has not discouraged that the brace can be used as an option of early intervention in AIS for the smaller thoracic and thoracolumbar curves with a CA between 15° and 25°. However, to substantiate this, additional studies are needed such as a stronger prospective randomized design with a control group and a larger number of subjects [13]. These studies should entail quantifiable monitoring of brace compliance, a more detailed evaluation for skeletal maturity, and a more detailed stratification of outcome than using the initial Risser stage.

## 5. Conclusions

In conclusion, this prospective observational study demonstrated an immediate in-brace correction of the frontal deformity of AIS in six subjects. In follow-up, there was an immediate correction of one-fourth of the deformity. At skeletal maturity, the deformities improved more than expected when compared to that of the natural history/observation and similar to that of other soft braces. No long-term deformity progression was seen.

## Figures and Tables

**Figure 1 jcm-11-00264-f001:**
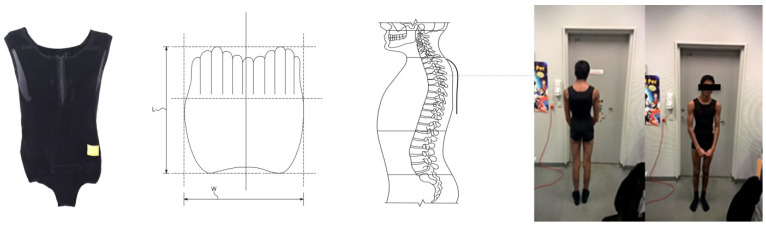
The ‘suit’ of the Spinaposture brace (**left**), a schematic diagram of the ’shield with fingers’ and placement of the shield (**middle**) and the Spinaposture brace© worn by a subject (**right**).

**Figure 2 jcm-11-00264-f002:**
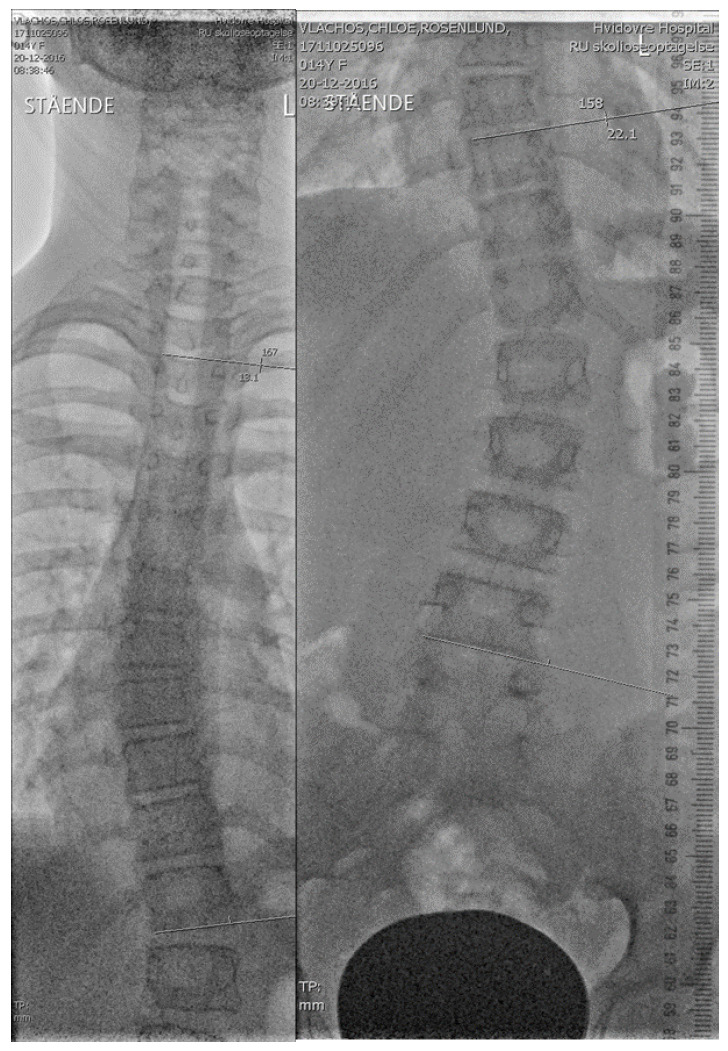
An example of radiographs using the low dose technique.

**Figure 3 jcm-11-00264-f003:**
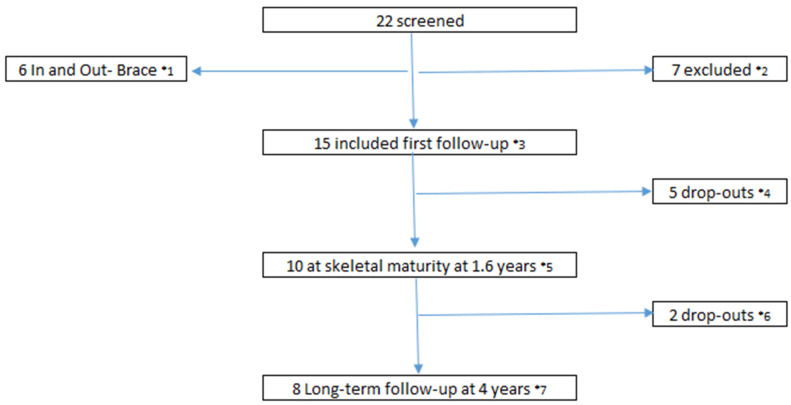
Timeline of the subject’s participation. *^1^ Six patients had initial in and out-of-brace radiographs as part of the inclusion process. *^2^ Seven patients did not fulfil the inclusion and exclusion criteria, thus were excluded. *^3^ Fifteen subjects were included with an improvement of one-fourth in Cobb’s angle. *^4^ Five subjects dropped out due to change to other brace strategies by their own accord. *^5^ Evaluation at skeletal maturity and when the soft brace was discontinued. *^6^ Two subjects did not participate in the follow-up at four years. *^7^ Evaluation at follow-up at four years.

**Figure 4 jcm-11-00264-f004:**
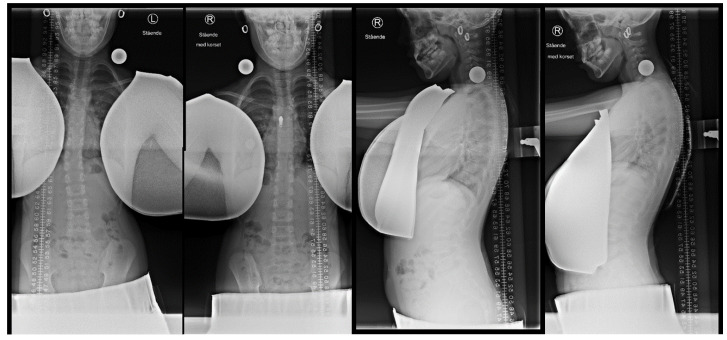
In-brace (picture two and four from left) and out-of-brace (picture one and three from left) of the frontal and sagittal radiographic examinations of a 10-year-old girl. The brace can be identified by the zipper at chest level in picture two and the ‘fingers’ are seen in picture four.

**Figure 5 jcm-11-00264-f005:**
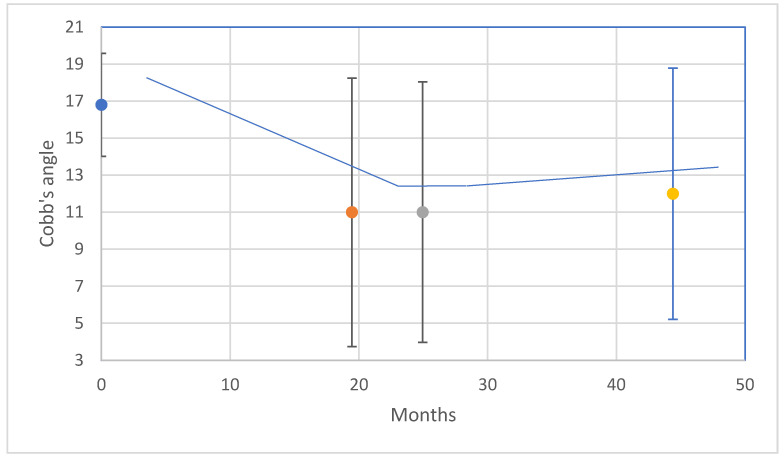
Average changes in Cobb’s angle (±one standard deviation) over time in the follow-up from the brace initiation to the long-term follow-up. (Upper-A).

**Figure 6 jcm-11-00264-f006:**
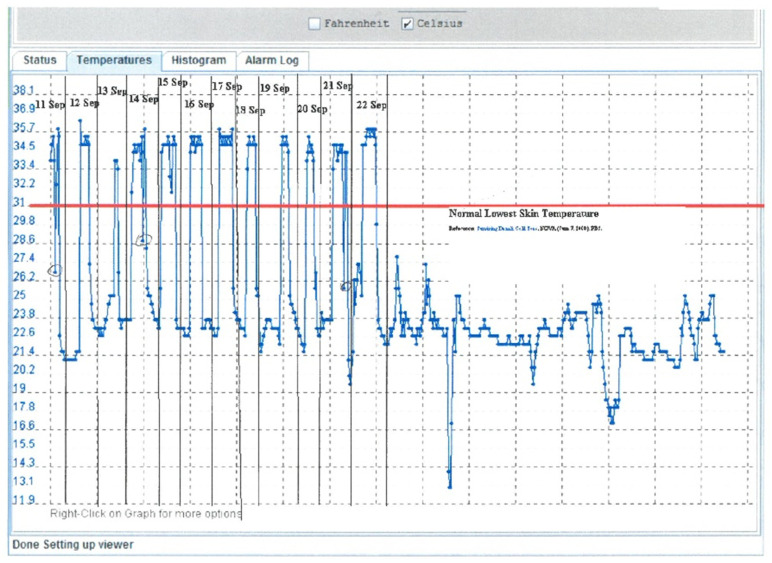
An example of a period with daytime and nighttime use of the brace. The heat sensor would register the temperature rise when the brace was worn. A period of brace use is seen on the left-hand side of the figure (compliance) and a period of non-compliance is seen on the right-hand side of the figure. (Upper-A).

**Table 1 jcm-11-00264-t001:** Patient Characteristics. *^1^ Patient ID. *^2^ Age when diagnosed (years). *^3^ Gender (fem = female, male = male). *^4^ Age of menarche (years). *^5^ Not relevant since male. *^6^ Pre-menarche. *^7^ Type of scoliosis (C, single curve; ju, juvinil; de, dextrokonvex; si, sinistrokonvex; TS, thoracic; TLS, thoracolumbar; s-s, combined curve). *^8^ AIS related events. *^9^ MRI of columna. *^10^ intermittent back pain, which was relieved by the brace, and the brace was omitted temporarily at the beginning of 2017 since her spine was straight, but radiographic monitoring is continued. *^11^ anisomelia of 1.3 cm, but still had radiologic diagnosed AIS after leg length correction (after hip fracture). *^12^ back pain after using the Spinaposture brace and whilst using the Chêneau light brace. *^13^ normal MRI. *^14^ improvement at First Visit. *^15^ improvement at Skeletal Maturity. *^16^ improvement at Long-Term.

Pt *^1^	Age *^2^	Sex *^3^	Risser	Mena *^4^	Type of Scoliosis *^7^	AIS Events *^8^	MR col. *^9^	CA ^FV^ *^14^	CA ^SM^ *^15^	CA ^LT^ *^16^
1	14.6	male	2	- *^5^	high de-TS	-	-	9.6	6.85	9.1
2	9.1	fem	0	12.2	high de-TS	Back pain *^10^	MR ia *^13^	10.85	16.5	14.9
3	14.8	fem	1	15.7	s-s de-TLS	Anisomelia *^11^	-	1.3	3.7	0.65
4	13.9	male	1	- *^5^	de-TLS	-	-	7.4	−0.7	−2.3
5	10.9	fem	1	12	s-s de-TLS	-	-	1.45	4.95	6.05
6	15.2	male	2	- *^5^	low de-TLS	-	-	3.75	-	-
7	9.7	fem	0	pm *^6^	s-s de-TLS	-	-	4.2	4.4	−0.35
8	13.5	fem	0	14.4	s-s de-TLS	Back pain *^12^	MR ia *^13^	0.8	-	-
9	15.1	fem	1	15.3	low de-TS	-	-	2.45		-
10	14.6	fem	2	15.5	low de-TS	-	-	2.7	3.8	-
11	7.2	fem	0	pm *^6^	s-s ju-deTLS	-	-	3.5	1.4	-
12	12.8	fem	0	13.6	s-s de-TLS	-	-	4.45	−3.8	−0.1
13	11.5	fem	0	13	C si-TLS	-	-	2.2	−0.4	−3.25
14	7	male	0	- *^5^	C ju-deTLS	-	-	14.4	-	-
15	15.2	fem	2	14.8	s-s de-TLS	Back pain	MR *^13^	4.9	-	-

**Table 2 jcm-11-00264-t002:** The achieved and expected outcomes after using the soft brace at skeletal maturity. * expected outcome in accordance to Costa et al. (2021) [13] (Lower-B).

		Achieved in the Present Study			Expected Number of “Improved/Straight” or “Stable” *
Risser	N	Improved/Straight	Stable	Improved/Straight or Stable	
0–1	11	5	6	11	6 (42%)
0–2	15	6	9	15	9 (71%)

**Table 3 jcm-11-00264-t003:** Secondary radiographic parameters. *^1^ Patient ID. *^2^ Time in the brace (in months). *^3^ changes in descriptive morphology/classification (straight = developed from initial curve straight, → = changed to, otherwise unchanged/Type of scoliosis (C, single curve; ju, juvenile; de, dextrokonvex; si, sinistrokonvex; TS, thoracic; TLS, thoracolumbar; s-s, combined curve). *^4^ Level of apex vertebrae for the primary curve (→ = changed to, otherwise unchanged). *^5^ Difference in Metha angle at apex vertebra. *^6^ change in Nash and Moe’s classification at apex vertebra for the primary curve (→ = changed to, otherwise unchanged).

Pt *^1^	Time in Brace *^2^	DM/C *^3^	A.Ver. *^4^	Metha.A. *^5^	NashMoe *^6^
1	12.3	→straight	Th3→Th12	13.4	0→0
2	36.6	→straight	Th4→Th5	5.9	0→0
3	28.8	→straight	Th5	6.2	0→0
4	13.7	deTLS	Th12→Th11	14.9	0→0
5	22.3	→straight	Th12→Th11	11.4	0→0
6	6.8	low de-TLS	Th8→Th10	4.5	0→0
7	24.7	→straigh	Th9	1.5	0→0
8	8.3	s-s TL	Th9	7.3	1→1
9	7.1	low dTS	Th10	11	0→1
10	16.7	low dTS	Th9	11.3	0→0
11	12.6	→straight	Th9→Th10	19.5	0→0
12	12.8	s-s de-TLS	Th12	10.4	0→0
13	34	C si-TLS	Th10	21.2	0→1
14	4.5	C ju-deTLS	Th12→Th11	27.8	0→1
15	8.1	s-s de-TLS	Th10	13.4	0→0

**Table 4 jcm-11-00264-t004:** Brace events and brace history. *^1^ Patient ID. *^2^ Brace events and clinical comments. *^3^ Types of physiotherapy (- = ordinary strengthening spine muscle exercises). *^4^ omitted the brace during periods due to hot weather and had a subsequent progression in CA, which regressed again using the brace. *^5^ irritation in the crotch, which was amended partly by modification of the brace. *^6^ intermittent back pain, which was relieved by the brace, and the brace was omitted temporarily at the beginning of 2017 since her spine was straight, but radiographic monitoring is continued. *^7^ back pain after using the Spinaposture brace and whilst using the Chêneau light brace. *^8^ change brace to Chêneau light. *^9^ change brace to providence in a short period and afterwards Chêneau light. *^10^ Excluded. *^11^ Straight spine and the brace was omitted. (Lower-B).

Pt *^1^	Brace events *^2^	Brace Change	Exercise *^3^
1	-	-	schroth
2	Sum *^4^ & Irr *^5^ & BP *^6^	-	schroth
3	Sum *^4^ & Irr *^5^	-	schroth
4	-	-	-
5	-	-	-
6	E *^10^	-	-
7	-	-	-
8	BP *^7^ & E *^10^	chea *^8^	schroth
9	Irr *^5^ &E *^10^	prov/chea *^9^	schroth
10	-	-	schroth
11	OM *^11^	-	-
12	-	-	-
13	-	-	-
14	E *^10^ & OM *^11^	-	Roolfing
15	E *^10^	-	-

## Data Availability

Data from this study can be requested from the corresponding author.

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
