# Peer review of "Evaluation of Brace Treatment Using the Soft Brace Spinaposture: A Four-Years Follow-Up"

_jcm, 2022, doi:10.3390/jcm11010264_

Round 1

Reviewer 1 Report

Interesting work introducing a novel form of bracing for AIS.

It would be interesting to have a more detailed function of the brace and how this can be utilized in order to obtain the most beneficial results.

It was not very clear reading the manuscript whether measurements were made from xrays in or out of brace, and how long was the patient out of brace before xrays. How long was the brace discontinued until final evaluation.

15 degrees for initiation of treatment appears a little in the low side.

The drop out rate with only 6 patients completing the study out of originally 22 seems quite sinificant for obtaining meaningful results.

There is no study group for comparison. The authors rely on cohort groups from the literature and statistical extrapolations in order to reach conclusio

Author Response

Dear Reviewer

Thank you for an excellent summary and most relevant comments to the article ' Evaluation of brace treatment using the soft brace Spinaposture. A four-year follow-up.'

We have revised the manuscript according to your comments and will describe this in detail. Please see the enclosed word file.

Best regards

Reviewer 2 Report

This manuscript is about a newly-developed soft brace for AIS patients with relatively small Cobb's angle (CA). Although the development of a more comfortable brace with as-high treatment effect as existing braces seems important and to mean much for patients, there are several issues to be resolved.

#1 Page 2, lines 45–49 and Figure 1.

Although the concept of the Spinaposture ––to induce adequate thoracic kyphosis––

was easy to understand, it was not easily understandable how it was done. Especially, the sentence "The brace does not induce frontal plane correction." was confusing. It DID induce frontal plane correction, as shown in the results of immediate in-brace correction, didn't it? In addition, the right 2 pictures were too small to distinguish the brace from clothes. Please consider presenting photos of the brace alone instead of that of prototypes.

#2 Page 2 line 75 – page 3 line 85.

The inclusion/exclusion criteria and the study protocol did not seem clearly described. Was the prior brace treatment one of the exclusion criteria as written in page 4, lines 148–150? Were all patients that met the inclusion criteria and did not meet the exclusion criteria included in the study as long as the informed consent was obtained? If the correction of the deformity in the frontal plane in three months was less than one-fourth, was the brace discontinued? If so, how many patients met the criterion?

#3 It is my understanding that the biggest feature of the Spinaposture is to induce the thoracic kyphosis. If so, why was the sagittal radiograph not evaluated throughout the study?

#4 Page 3, lines 110–114.

In general, the direct comparison with patients in other studies are not appropriate. The results in previous reports can be referred but they cannot be used as the target for the direct comparison. Were the lines 112–114 meant to summarize the reference #26 and #27? Please describe clearly what was written.

#5 Page 4, lines 115–116.

It is unclear what "multiple comparisons" means. The authors should state what was compared with what.

#6 Page 4, lines 117–125.

Secondary parameters (outcomes?) seem too many to be written in the Results section. If the authors intended to include all the parameters written in this paragraph as secondary outcomes, all the results regarding those outcomes should be written in the Results section.

#7 Page 4, lines 125–126.

"We performed the non-parametric binominal analyses for 125 the primary and secondary parameters at skeletal maturity."

I do not understand the sentence fully. Please explain.

#8 Page 4, lines 126–130.

Did the authors perform multiple linear regression analysis with the change in CA as a dependent variable and with all other variables as independent variables included in one model? If so, it is inappropriate, for the model includes too many independent variables. In general, the sample size required for the linear regression analysis are considered to be 15-times of the number of independent variables included. In the present study, only 10 patients were followed-up with radiographs at skeletal maturity, which indicate that even simple linear regression analysis would be inappropriate.

#9 Page 4, lines 147–158 and Figure 3.

The flow of patients' inclusion, exclusion, and drop-out is difficult to understand even with intensive reading. Please consider revising. At least, some of the word "excluded" in Figure 3 would need to be replaced by "drop-out".

#10 Tables 1 and 2.

The letter size in the Tables legends is too small to read. It seems to include unnecessary information to understand the present study. Please focus on important information and make the letter size big.

#11 Page 6, line 188.

The statistical method is unclear. What kind of analysis gave the p value of <0.001?

#12 Figure 6.

I do not understand what the authors intended to show in Figure 6, probably due the low resolution of the figure.

Author Response

(The authors gave the same response as above.)

Round 2

Reviewer 1 Report

I believe the authors have peformed changes as requested by the reviewers to their capacity with the limitations posed by the study design.

Author Response

Dear reviewer

Thank you for this evaluation.

Kind regards,

Christian

Reviewer 2 Report

Thank you for the vigorous revising and kindly answering all my questions. I believe that the manuscript is much better now. I am almost satisfied with the answers and the revised manuscript, however, there is one more question. Please refer to the Word file attached.

Author Response

Thank you for the comments for the first revisions as well as for the ones to this second revision. We acknowledge that this has improved the quality of the manuscript. We thank you.

We have modified the manuscript in accordance  with the reviewer good comments to the second revision. Thank you!
